# Neither influence nor selection: Examining co-evolution of political orientation and social networks in the NetSense and NetHealth studies

Cheng Wang[1]*, Omar Lizardo[2], David S. Hachen[3]

1 Department of Sociology, Wayne State University, Detroit, MI, United States of America, 2 Department of Sociology, University of California Los Angeles, Los Angeles, CA, United States of America, 3 Department of Sociology, University of Notre Dame, Notre Dame, IN, United States of America

* chengwang@wayne.edu

**Data Availability Statement:** The NetSense data is attached as S1 File. The NetHealth data are now publicly online at http://sites.nd.edu/nethealth.

## Abstract

Political orientation is one of the most important and consequential individual attributes studied by social scientists. Yet, we know relatively little about the temporal evolution of political orientation, especially at periods in the life course during which individuals are forming new social relationships and transitioning to new relational contexts. Here we use Stochastic Actor-Oriented models (SAOMs) to examine the co-evolution of political orientation and social networks using two feature-rich, temporal network datasets from samples of students making the transition to college at the University of Notre Dame (i.e. the *NetSense* and *NetHealth* studies). Overall, we find a great deal of stability in political orientation, with a slight tendency for the 2011 NetSense study participants to become more conservative during their first four semesters in college, but not the 2015 NetHealth study participants. Partisanship is the best predictor of changes in political orientation, with students who identify or vote as Republicans becoming more conservative over time. Neither network influence nor selection processes seem to be driving observed changes. During this formative period, relatively stable identities such as party affiliation predict changes in political orientation independently of local network dynamics, selection processes, socio-demographic traits, and dispositional factors.

## Introduction

An individual's political orientation reflects his or her relative position or standing on a liberal-conservative (left-right) spectrum [1–3]. In a classic statement, the political scientist Philip Converse theorizes that political orientation, as reflective of ideology, should have a generalized effect on a variety of stances across a wide range of issues and policy domains, including the economy, welfare policy, the propriety of foreign intervention, ideas about the role and size of government, and racial and gender attitudes, among others [1]. Consistent with this hypothesis, researchers find political orientation to be a key predictor of gender role ideology

**Funding:** This work was supported by National Institutes of Health #1 R01 HL117757-01A1. The funders had no role in study design, data collection and analysis, decision to publish, or preparation of the manuscript.

**Competing interests:** The authors have declared that no competing interests exist.

[4–6], racial perceptions and preferences [2], favoritism of higher-status groups vs. lower-status groups [7], and social and economic equality policies such as affirmative-action programs, social security, social welfare programs, and Equal Rights Amendment (ERA) [8, 9].

To date, the bulk of research on political orientation in the United States concentrates on its socio-demographic correlates. For instance, we know older adults tend to place themselves toward the conservative side of younger adults [6], but differences between men and women are small [6, 10]. Substantively large differences in political orientation based on racial identification have emerged in recent decades, with whites placing themselves to the right of ethnocidal minorities [3]. Religious preference has complex effects on political orientation in the United States. Mainline Protestants tend to be less conservative than those who identify as "born again" [11]. The majority of Catholics, like most Americans, are not consistently liberal or conservative on all salient political issues and unlikely to place them on the extremely liberal position [12], although they are internally diverse in terms of orientation [13]. In addition, although individuals who report being more generally religious are also more conservative than those who are less religious, this association only holds among those who are active in politics [14].

Despite its significance for a variety of social-scientific outcomes and its systematic linkages to socio-demographic characteristics, we know relatively little about the origins and temporal evolution of political orientation, especially its linkages to peer influence processes in social networks. There is a dearth of research on the co-evolution of political orientation and network ties, especially as it pertains to periods in the life course, such as adolescents going to college [4], during which people form social relationships as they transition to new contexts, resulting in turnover and re-organization of discussion and political networks [15, 16].

More recently, political network researchers have become interested in the question of the co-evolution of political orientation and social networks, although only one study to date has analyzed empirical data germane to this question [17]. As such, while we know quite a lot about the socio-demographic correlates of political-orientation, especially in the cross-section, we are only beginning to better understand the linkage between political orientation and social network processes of influence, sorting, and tie-formation. The current study makes headway on this front.

## Dynamics of selection and influence

From the network science perspective, lots of human traits, behaviors, opinions, and attitudes are found to cluster non-randomly in social networks [18], with some arguing that this clustering is due to direct person-to-person influence processes. These include obesity [19, 20], depressive symptoms [21], academic achievement [22], physical activity [23], delinquent behavior [24], substance use [25–27], religious preference [28], and musical tastes [29]. Such homogenizing network processes can be driven by social learning [30], peer pressure [31], persuasion [32], punishment for nonconformity [33], and social control [34].

This previous work is particularly relevant to the study of the dynamic evolution of political orientation in social networks. Social ties and political orientation can evolve in tandem via a variety of processes. In a peer group influence process, people form ties irrespective of their political orientations, but later on adjust their orientation to come closer to that of the people they connect to, so that an individual's political orientation becomes a moving average reflecting the orientations of the people they are connected to in the network [15]. Work done from the point of view of "social contagion" theory [19, 20, 35, 36] presumes that such peer group influence and role-modeling processes are the primary sources of non-random clustering of traits in social networks.

There are two other processes, not based on influence mechanisms, that may lead to dyads in networks being more likely than chance to share the same trait. One, a positive selection process, is when individuals who already agree in political orientation self-select into forming a tie. Here the tie is the product of the previous agreement and not the other way around [37–39]. A negative selection process, in which ties among individuals who disagree in political orientation are more likely to decay, will also result in non-randomly clustering of political orientation in social networks [40]. The only work that has attempted to disentangle these processes in the case of political orientation, found evidence consistent with social influence over selection processes, showing that people tend to adopt the political orientation of their associates, but finding little evidence that either conservatives or liberals tend to choose like-minded friends [17].

## Political discussion networks

A now well-established line of work looks at the role of "political discussion networks" as either mitigating or amplifying existing political orientation among people. From this perspective, increasing largely homogeneous discussion networks function as "echo chambers," essentially amplifying existing opinions and attitudes and preventing people from being exposed to countervailing views [41]. People trapped in homogeneous discussion networks may become either more liberal or more conservative than people who discuss politics with a more diverse set of others and are thus exposed to countervailing opinions [42]. Echo chamber dynamics at the level of political discussion networks can also be amplified by a "selective disclosure" mechanism. This refers to the tendency of people to withhold dissenting opinions from others with whom they disagree, creating the *appearance* of highly homogeneous local social contexts in terms of political orientation, despite the existence of latent heterogeneity [43].

## Partisanship and polarization

There has been a debate about whether Americans are segmented and polarized along partisan lines. Political scientists Alan Abramowitz and Kyle Saunders claim that the divisions between Republicans and Democrats are deep and now involve the great bulk of the US population. They point out that partisan polarization has increased significantly over time, with the gap between Republicans and Democrats being more than twice as large in 2004 than it was in 1972. Additionally, the correlation between partisanship and political orientation has increased dramatically since 1972 and especially since 1988 [44]. In the same way, analysts such as Bobbio [9] suggest that while the political world of large-scale societies is getting increasingly complex, political ideology and the left/right division continues to be a robust line of differentiation structuring the preferences and values of large swaths of the population. Overall, from this perspective, the division in worldviews between liberals and conservatives is starker now than it has ever been, and may have begun to spill over into previously "non-political" views and attitudes [45].

Other scholars concede the objective existence of partisan divisions but point out that they do not exercise systematic influence on ideology or political behavior. Baldassarri and Gelman [45], examining a number of attitudinal domains, show that while the correlation between political orientation and attitudes does increase over time, the inter-correlation between attitude issues remains flat (a classic measure used by Converse to index ideological "constraint" [1]). They point out that we may live in a world of "partisans without constraint". In the same way, Fiorina, Abrams, and Pope [46] argue that the conception of the mass public as deeply polarized around extreme attitude positions, and thus engaged in a protracted "culture war," is empirically inaccurate. Instead, they show that most Americans are moderate in their political

orientations and hold cross-cutting attitudes mixing liberal and conservative positions across a wide variety of domains.

Importantly, previous work shows that partisanship, in the form of party affiliation, is a robust predictor of political orientation. Democrats tend to identify as liberals and Republicans as conservatives [1, 44], and the correlation between one's voting decision on party affiliated candidates and political orientation is high, and in one study, above .90 [47]. However, it is unclear whether strong (or weak) partisans are more or less likely to be susceptible to change their political views in the context of a major life transition. We will address that question in the analysis that follows.

## Contributions of the current study

This study contributes to the research on political orientation in three significant ways. First, we include traditionally considered socio-demographic, cognitive factors, and social network determinants of political orientation within a single statistical modeling framework. Such an approach allows us to ascertain their cross-sectional and temporal effects on the evolution of political-orientation at a critical period.

Second, we use rich longitudinal data over four semesters from two separate cohorts of undergraduate students at the University of Notre Dame. In so doing we can ascertain how changes in political orientation are affected in systematic ways as individuals make significant life course transition in a context.

Third, we introduce a robust statistical analysis method from social network analysis, the Stochastic Actor-Oriented modeling (SAOM) strategy, to the political orientation literature. In this respect, our work builds on but goes beyond Lazer et al., who first explored how political attitudes and affiliations coevolve in social networks [17]. That work used a less efficient multi-step strategy, examining political orientation dynamics using ordered logit regression models and network dynamics using multiple regression quadratic assignment procedure (MRQAP) and exponential random graph models (ERGMs). A key advantage of the SAOM approach implemented here is its capability to simultaneously estimate the effects of social influence, selection, socio-demographic, and temporal trends effects in one step [48, 49]. This allows us to consider the co-evolution of political orientation and social networks in a statistically efficient way, while accounting for temporal autocorrelation among the different mechanisms and guarding against likely sources of spuriousness.

## Materials and methods

### Data

The data used in this study come from NetSense [50] and NetHealth [51, 52] studies, both of which were approved by the Institutional Review Board at the University of Notre Dame. Each study randomly selected a cohort of first-year undergraduate students at Notre Dame and written informed consents were obtained from all participants. The NetSense study recruited 196 participants in August 2011. The NetHealth study recruited its first tier of 387 participants in August 2015 and the second tier of 96 participants in October 2015, which add up to 483 participants in our sample. Each participant was asked to complete an entry survey when joining the study, and information was collected on his or her gender, racial identification, religious preference, residence hall on campus, parental income and education, and personality.

The follow-up surveys were conducted during the spring semester in their first year, the fall semester in their second year, and the spring semester in their second year at Notre Dame. (The NetHealth study also recruited a third tier of 209 participants in March 2016. We exclude them from our sample so that each NetHealth study participant has four waves of data as

NetSense study participants.) The response rates of the four consecutive surveys (with the entry survey included) are 100%, 91%, 89%, and 77% for NetSense study participants, and 100%, 91%, 83%, and 80% for NetHealth study participants, respectively. In the entry survey and three follow-up surveys, study participants reported their positions on the liberal-conservative spectrum and the frequencies of discussing politics with their friends, and NetHealth study participants disclosed whether they regarded themselves as Republicans, Democrats, or Independents. Given the high stability of partisanship status among the NetHealth study participants, we treat it as a time-constant variable. With regard to the NetSense study, unfortunately, the same partisanship question was not included in the survey. Instead, NetSense study participants were asked whether they voted for Mitt Romney, Barack Obama, or other candidates during the presidential election of 2012 in the last of the three follow-up surveys. We use this measure as a proxy for partisan affiliation.

We aggregate the network ties among the participants in each semester using the call detailed records (CDRs) obtained from their smartphones, although the specific operations are a little bit different across the two studies. In the NetSense study, each participant was provided with a free Android-powered smartphone and phone plan for two years (including unlimited voice calls to and from other mobile phones, 200 voice-call minutes per month to and from other landlines, unlimited texting, and unlimited data from August 2011 to May 2013). These smartphones were pre-programmed to automatically back up the communication event data (i.e., text messages, voice calls, emails, and Facebook posts), and thus theoretically there were no missing network data. NetHealth study participants, however, brought their own smartphones and used their own phone plans. When they participated in the study they agreed to install an app devised by the study team (which served for the same purpose of backing up the communication event data as in the NetSense study) and kept it running in the background. In practice there were missing network data for 65 participants because some of them never installed the app, some had phones with limited Read Only Memories (ROMs) and data backups were technically impossible, and a few had Windows phones which were not compatible with the app. These missing network data are labeled so that the analytical model knows their existence and prepares to deal with them.

The raw CDRs are used to construct an adjacency matrix representing a communication network (based on text and calls) for each semester. The adjacency matrix was built based on communication events in the semester of the corresponding survey from which we obtained political orientation data was administered. A directed tie between person $i$ and person $j$ exists in a given semester if $i$ initiated a communication event (call or text) with $j$ during that semester. Each adjacency matrix is asymmetric above and below the diagonal, given that communication relationships between an "ego" (i.e. a focal study participant) and his or her "alter" (i.e. another participant to whom ego is directly connected) is a directed tie and might not always be reciprocal. Since these network ties are between with-study participants, we do not use a cutoff on tie strength, i.e., the frequency or duration of communicant events.

## Measures

The political orientation is measured as a 7-point Likert scale that ranges from "extremely liberal" (1) to "extremely conservative" (7). Demographic factors include gender (0 = Men, 1 = Women) and racial identification (0 = White, 1 = Latino, 2 = African American, 3 = Asian American, 4 = Other race). Religious preferences are classified into Catholic (0), Protestant (1), other religion (2), and no religion (3). Socioeconomic status (SES) is measured using parental annual income (1 = Less than 25k, 2 = 25~50k, 3 = 50~75k, 4 = 75~100k, 5 = 100~150k, 6 = 150~200k, 7 = 200~250k, 8 = 250k and more) and parental highest degree

(1 = Not graduate from high school, 2 = High school graduate, 3 = Postsecondary school or some college, 4 = College/University degree, 5 = Graduate or professional degree). Cognitive factors consist of the big five factors in personality trait ratings [53] which include extraversion (a standardized factor score of 8 items; Cronbach's α = 0.88 and 0.87 among NetSense and NetHealth study participants, respectively), agreeableness (a standardized factor score of 9 items; Cronbach's α = 0.78 and 0.80 among NetSense and NetHealth study participants, respectively), conscientiousness (a standardized factor score of 9 items; Cronbach's α = 0.78 and 0.83 among NetSense and NetHealth study participants, respectively), neuroticism (a standardized factor score of 8 items; Cronbach's α = 0.79 and 0.82 among NetSense and NetHealth study participants, respectively), and openness (a standardized factor score of 10 items; Cronbach's α = 0.79 and 0.79 among NetSense and NetHealth study participants, respectively). Political factors include the frequency of discussing politics with friends (0 = Not at all, 1 = Less than 1~2 times a month, 2 = 1~2 times a month, 3 = 1~2 time a week, 4 = Three times a week or more) and partisanship (0 = Republican or NetSense study participant who voted for Mitt Romney during the presidential election of 2012, 1 = Democrat or NetSense study participant who voted for Barack Obama during the presidential election of 2012, 2 = Independent or NetSense study participant who voted for other candidates during the presidential election of 2012). Finally, the data also contains information on each participant's residence hall. There are currently 30 residence halls on the campus of Notre Dame and one of them established after NetHealth study participants enrolled in fall 2015. Therefore, NetSense and NetHealth study participants were distributed into 29 residence halls. Given participants living in the same residence halls could be more likely to establish relationships among themselves than with those living in different residence halls, we use the commonality of their residence halls to predict the formation and maintenance of social network ties among them.

## Method

In the present study, we apply the SAOM strategy to examine the co-evolution of political orientation and social networks implemented in the R-based Simulation Investigation for Empirical Network Analysis (*RSiena*) software package [54]. The prototype of SAOM strategy was advanced by Snijders in 2001 but at that time the changing network was the only outcome variable in the model [55]. Its mature form was elaborated in Snijders et al. [48] and Steglich et al. [49], and has gradually turned into one of the dominant statistical techniques in analyzing the interdependent changes in both human behaviors and social networks. Since 2012 there are more than 40 peer-reviewed papers adopting the SAOM strategy published every year. The SAOM strategy has been widely employed by researchers in public health, biological science, medical science, business and economics, psychology, sociology, and political science.

The SAOM strategy assumes that each individual (or actor) will make decisions optimizing his or her political orientation and network status in the next time step based on an evaluation of his or her current state of political orientation-network tie configuration. The evaluation function of each individual $i$ is defined as $f_i(\beta,x) = \sum_k \beta_k s_{ik}(x)$, where $\beta_k$ is the estimated parameter for the $k$th actor-specific effect $s_{ik}(x)$. The choice of each individual is conceptualized to follow a stochastic conditional logit model with regard to the change in utility before and after making the choice. The SAOM strategy adopts a Method of moments (MoM) estimation to estimate the parameters such that the main characteristics of the political orientation and network ties are most closely approximated. Therefore, the underlying estimation procedure of the SAOM strategy is different from that of logistic regression. However, the interpretation of the log-odds ratio estimate, as well as odds ratio, is the same as in logistic regression. Let's suppose the individual's current state is $x_a$ and he or she decides to change to $x_b$ at the next time

point. The log odds ratio of the utility change is thus $f_i(\beta,x_b)-f_i(\beta,x_a)$ and the probability ratio of change is $\exp[f_i(\beta,x_b)-f_i(\beta,x_a)]$. A positive value of the log odds ratio indicates the preferred direction of change, while a negative value indicates that people avoid such a change. In the SAOM framework, the evaluation functions of changes in political orientation and network ties are estimated simultaneously to generate a set of interdependent equations with the rate functions $\lambda_i(\alpha,x)$, which indicate the expected frequency of changes in the political orientation or network ties the individual makes between observation points. The model is then estimated by simulating the networks and behavior forward in time. Therefore, there are many micro-steps in which actors update their evaluation functions by changing their states on political orientation and network tie choice (i.e., -1 unit, no change, or +1 unit).

We handle missing data on the outcome, predictors, and network ties using the strategy suggested by Huisman and Steglich [56] and Ripley et al. [54]. Model convergence is checked by using criteria of both *t* statistics for deviations from observed statistics (i.e., ideally less than 0.10 for each parameter) and the overall maximum convergence ratio (i.e., ideally less than 0.25). We assess the goodness of fit of estimated SAO model for each cohort by performing the Monte Carlo test of Mahalanobis Distance statistics [57] in terms of out-degree distribution, in-degree distribution, geodesic distance distribution, and triad census. The *p*-value in each test is greater than 0.05, suggesting that our estimated SAO models are adequate in reproducing the observed network statistics. Since we have four waves of data in political orientation and network ties, time heterogeneity is tested for estimated parameters and the results show that the co-evolution of one's political orientation and network ties was not significantly different across any two consecutive semesters.

## Sample attrition

Sample attrition is a typical problem for longitudinal studies. S1 Table displays the summary statistics of gender, racial identification, and religious preference among NetSense and NetHealth study participants, which are generally similar between stayers and dropouts from wave 2 to wave 4. To address sample selection concerns, we also ran additional logistic regression models predicting the dropout status of NetSense and NetHealth study participants with these factors. The only statistically significant effect is that a NetHealth study participant identifying as the "other" race is 127% more likely to drop out of the study than a white participant at wave 4 ($p < 0.05$). Overall, attrition bias based on the usual demographic factors appears to not be a concern in the current study.

## Results

### Summary statistics

Both the NetSense and NetHealth data include an additional option of "Not sure" and allowed their participants to voluntarily opt out of the political orientation question. In practice, no participants in our sample chose that option or refused to answer this question. Therefore, consistent with previous work [47], a great majority of college students can place themselves in the liberal-conservative continuum. Political orientation can take values ranging from 1 (extremely liberal) to 7 (extremely conservative). As shown in Table 1, both NetSense and NetHealth study participants had an average placement around 4 (moderate). The frequency distribution suggests that very few participants self-reported extreme stands on either side of the liberal-conservative spectrum. This is consistent with previous work showing almost half people clustering within one unit of the measure's mid-point [44, 46].

In Table 1 we also report the network autocorrelation measure Moran's *I* which assess the degree of clustering along political orientation dimension within the social network at each

**Table 1. Frequency distribution, mean/SD, and network autocorrelation of political orientation among NetSense and NetHealth study participants.**

|  |  | Wave 1 | Wave 2 | Wave 3 | Wave 4 |
|---|---|---:|---:|---:|---:|
| NetSense | Extremely liberal | 1% | 0% | 2% | 1% |
|  | Liberal | 12% | 13% | 13% | 15% |
|  | Slightly liberal | 16% | 17% | 20% | 15% |
|  | Moderate | 19% | 25% | 18% | 21% |
|  | Slightly conservative | 25% | 20% | 24% | 23% |
|  | Conservative | 24% | 22% | 21% | 22% |
|  | Extremely conservative | 3% | 3% | 2% | 3% |
|  | Mean (SD) | 4.36 (1.43) | 4.29 (1.41) | 4.21 (1.46) | 4.29 (1.48) |
|  | Moran's I | 0.09 | 0.08 | 0.14 | 0.10 |
|  | n | 196 | 178 | 174 | 151 |
| NetHealth | Extremely liberal | 3% | 2% | 1% | 2% |
|  | Liberal | 16% | 20% | 23% | 26% |
|  | Slightly liberal | 17% | 15% | 17% | 18% |
|  | Moderate | 19% | 21% | 19% | 15% |
|  | Slightly conservative | 17% | 15% | 16% | 15% |
|  | Conservative | 25% | 25% | 22% | 22% |
|  | Extremely conservative | 3% | 2% | 2% | 2% |
|  | Mean (SD) | 4.20 (1.58) | 4.14 (1.56) | 4.01 (1.56) | 3.92 (1.61) |
|  | Moran's I | 0.07 | 0.10 | 0.12 | 0.12 |
|  | n | 483 | 440 | 401 | 386 |

wave. While Moran's *I* fluctuated over the four semesters among NetSense study participants and increased among NetHealth study participants, its positive values around 0.1 suggest that there was a slight clustering of political self-placement in the networks but not much.

We ran a series of supplementary analyses, three findings from which are worth reporting. First, period-specific means on political orientation did not differ statistically over any two consecutive semesters for either NetSense or NetHealth study participants. Second, group-specific means on political orientation did not differ across NetSense and NetHealth study participants at the first three waves. Finally, NetSense study participants were on average slightly more conservative than NetHealth study participants during the spring semester of their second year at Notre Dame, the last observation set ($p = .007$).

While there is overall a lot of stability in political orientation in each sample over time, we can still observe some non-trivial within-individual variation in self-placement over time, thus justifying the longitudinal analysis. Relevant descriptives are shown in Table 2. A majority of NetSense and NetHealth study participants (except wave 2 to wave 3 of NetHealth) maintained their same orientation positions across four semesters and over 90% did not change or changed one unit on the liberal-conservative spectrum. A small number of participants changed 2 units across two consecutive semesters and very few participants changed 3 to 4 units in their political orientation. Thus, we see a combination of fluctuation in political self-placement along with strong temporal correlations between consecutive placements.

The summary statistics for demographic, religious, socioeconomic, and personality factors are presented in Table 3. Almost half of the participants are women in each sample. A majority of participants are whites and Catholics, which is representative of the institution. Reflecting the socio-economic background of most students in the school, parents of over half of the NetSense study participants and over 70% of the NetHealth study participants earned more than 100k per year and over 80% of parents had a college/university degree or above.

**Table 2. Changing units in political orientation among NetSense and NetHealth study participants.**

|  |  | Wave 1 → Wave 2 | Wave 2 → Wave 3 | Wave 3 → Wave 4 |
|---|---|---|---|---|
| NetSense | -3 units | 0% | 0% | 1% |
|  | -2 units | 3% | 3% | 1% |
|  | -1 unit | 22% | 24% | 13% |
|  | 0 unit | 53% | 60% | 59% |
|  | +1 unit | 19% | 12% | 25% |
|  | +2 units | 3% | 1% | 1% |
|  | *n* | 178 | 159 | 132 |
| NetHealth | -4 units | 0% | 0% | 0% |
|  | -3 units | 1% | 1% | 0% |
|  | -2 units | 4% | 5% | 2% |
|  | -1 unit | 19% | 24% | 21% |
|  | 0 unit | 56% | 49% | 62% |
|  | +1 unit | 18% | 19% | 14% |
|  | +2 units | 2% | 2% | 1% |
|  | +3 units | 0% | 0% | 0% |
|  | +4 units | 0% | 0% | 0% |
|  | *n* | 440 | 367 | 312 |

Turning to political factors, as shown in Table 4, NetHealth study participants talked about politics with their friends somewhat more frequently than NetSense study participants. During the presidential election of 2012, there were around 7% more NetSense study participants who voted for Barack Obama than for Mitt Romney and only a few of them voted for other candidates. In contrast, many more NetHealth study participants declared themselves to be Independents, and those who considered themselves as Republicans and Democrats were about evenly distributed.

Network statistics are shown in Table 5. The number of outgoing ties among both NetSense and NetHealth study participants decreased over the four semesters, and stability in the personal network increased over time as indicated by the Jaccard index. The Jaccard index is the proportion of persisting ties divided by all ties existing at least in one of the consecutive waves. A value of 0.3 or greater is ideal to fit the gradual process of network evolution assumed in SAOMs [48]. This index increased from 0.41 to 0.50 among NetSense study participants and from 0.38 to 0.43 among NetHealth study participants. The proportion of reciprocal ties divided by all outgoing ties was 2% to 9% higher among NetSense study participants than among NetHealth study participants at each wave. The transitivity index which measures the proportion of 2-paths (ties existing between AB and AC) that are transitive (ties existing between AB, AC, and BC, which represent the dyadic relations among three participants A, B, and C), was also higher among the NetSense study participants than among NetHealth study participants.

## SAOM results

We estimate two SAO models, one for NetSense study participants and one for NetHealth study participants (Models 1 and 2 in Tables 6 and 7, respectively). Each model includes a set of two interdependent equations, one for predicting changes in political self-placement and the other for predicting changes in social ties in order to adjust for selection effects in which people with similar political orientation form ties and various local structural effects. Given the

**Table 3. Summary statistics of demographic, religious, socioeconomic, and cognitive factors among NetSense and NetHealth study participants.**

|  | NetSense (Wave 1) | NetHealth (Wave 1) |
|---|---|---|
| Women | 46% | 48% |
| Racial identification |  |  |
| White | 68% | 65% |
| Latino | 10% | 14% |
| African American | 6% | 6% |
| Asian American | 12% | 9% |
| Other race | 4% | 6% |
| Religious preference |  |  |
| Catholic | 69% | 74% |
| Protestant | 14% | 10% |
| Other religion | 2% | 5% |
| No religion | 15% | 11% |
| Parental annual income |  |  |
| Less than 25k | 7% | 5% |
| 25~50k | 12% | 7% |
| 50~75k | 15% | 9% |
| 75~100k | 14% | 8% |
| 100~150k | 26% | 20% |
| 150~200k | 7% | 11% |
| 200~250k | 7% | 11% |
| 250k and more | 12% | 29% |
| Parental highest degree |  |  |
| Not graduate from high school | 1% | 1% |
| High school graduate | 7% | 4% |
| Postsecondary school or some college | 7% | 5% |
| College/University degree | 32% | 31% |
| Graduate or professional degree | 53% | 59% |
| Big five personality traits: Mean (SD) |  |  |
| Extraversion | 0.00 (0.74) | 0.00 (0.72) |
| Agreeableness | -0.00 (0.60) | -0.00 (0.62) |
| Conscientiousness | 0.00 (0.60) | -0.00 (0.65) |
| Neuroticism | 0.00 (0.63) | -0.00 (0.66) |
| Openness | -0.00 (0.59) | 0.00 (0.59) |
| *n* | 196 | 483 |

long list of effects included, we present the political orientation equation and network equation of each model in two separate tables.

**Changes in political orientation.** We begin with the set of terms predicting changes in political orientation. As shown in Table 6, the rate parameters from the political orientation equation indicate that NetSense and NetHealth study participants had about one chance to change their positions in "microsteps" (i.e., -1 unit, no change, or +1 unit) on the liberal-conservative spectrum across consecutive semesters. The linear and quadratic shape parameters summarize the long-run distributional tendency in political self-placement occurring independently of actor attributes and network position. A positive linear shape parameter accompanied by a relatively smaller negative quadratic term indicates that there was a long-run growth tendency toward conservatism, but this growth tendency declined with the increasing conservative level of an individual. This is what we observe among NetSense study participants, as

**Table 4. Summary statistics of political factors among NetSense and NetHealth study participants.**

| | | Wave 1 | Wave 2 | Wave 3 | Wave 4 |
|---|---|---|---|---|---|
| NetSense | Frequency of discussing politics with friends | | | | |
| | Not at all | 6% | 10% | 5% | 13% |
| | Less than 1~2 times a month | 19% | 25% | 20% | 25% |
| | 1~2 times a month | 30% | 30% | 30% | 31% |
| | 1~2 time a week | 32% | 26% | 36% | 25% |
| | Three times a week or more | 13% | 9% | 9% | 6% |
| | Candidates voted in presidential election 2012 | | | | |
| | Mitt Romney | | | | 40% |
| | Barack Obama | | | | 47% |
| | Other candidates | | | | 13% |
| | *n* | 196 | 178 | 174 | 151 |
| NetHealth | Frequency of discussing politics with friends | | | | |
| | Not at all | 6% | 1% | 6% | 7% |
| | Less than 1~2 times a month | 14% | 11% | 7% | 12% |
| | 1~2 times a month | 30% | 22% | 18% | 29% |
| | 1~2 time a week | 31% | 34% | 39% | 34% |
| | Three times a week or more | 19% | 32% | 30% | 18% |
| | Partisanship | | | | |
| | Republican | 36% | 32% | 35% | 29% |
| | Democrat | 23% | 27% | 30% | 32% |
| | Independent | 41% | 41% | 35% | 39% |
| | *n* | 483 | 440 | 401 | 386 |

illustrated in Fig 1. But in the NetHealth data neither the linear shape effect nor the quadratic shape effect is statistically significant.

Beyond that, neither socio-demographic nor local network factors seem to be strong predictors of changes in political self-placement among either group of participants. Socio-demographic factors such as gender and racial identification, socioeconomic factors such as parental annual income and highest degree, network factors such as peer influence (measured as average similarity effect) and in-degree centrality, dispositional factors such as the big five personality traits, and interaction such as frequency of discussing politics with friends do not help predict changes in political orientation. The only exception is religious preference, with Net-Sense study participants who are in either the "other" or "no religion" categories less likely to become more conservative related to their Catholic and Protestant counterparts. Importantly, there is only a weak indication of the influence of a person's peers on changes in political orientation. The estimated parameter for the average similarity effect, i.e. ego's similarity to the average political orientation of peers, is positive but not significant in the NetSense or NetHealth data.

**Table 5. Summary statistics of social networks among NetSense and NetHealth study participants.**

| | NetSense (*n* = 196) | | | | NetHealth (*n* = 483) | | | |
|---|---|---|---|---|---|---|---|---|
| | Wave1 | Wave2 | Wave3 | Wave4 | Wave1 | Wave2 | Wave3 | Wave4 |
| Outgoing ties | 796 | 674 | 498 | 408 | 4,100 | 3,622 | 3,335 | 3,043 |
| Jaccard index | 0.41 | | 0.43 | 0.50 | 0.38 | | 0.38 | 0.43 |
| Reciprocity index | 0.93 | 0.92 | 0.92 | 0.89 | 0.84 | 0.87 | 0.86 | 0.87 |
| Transitivity index | 0.21 | 0.19 | 0.25 | 0.23 | 0.18 | 0.17 | 0.18 | 0.16 |

**Table 6. The political orientation equation in estimated SAO models for the NetSense and NetHealth study participants.**

| | NetSense | NetHealth |
| --- | --- | --- |
| | Model 1 | Model 2 |
| | beta (s.e.) | beta (s.e.) |
| Rate parameter (period 1) | 1.18*** (0.16) | 1.19*** (0.24) |
| Rate parameter (period 2) | 1.01*** (0.13) | 1.27*** (0.19) |
| Rate parameter (period 3) | 0.74*** (0.12) | 0.79*** (0.08) |
| Linear shape | 1.84** (0.63) | 0.31 (0.60) |
| Quadratic shape | -0.40*** (0.07) | -0.07 (0.08) |
| Woman | 0.37† (0.19) | -0.11 (0.20) |
| Latino | -0.08 (0.31) | 0.15 (0.18) |
| African American | -0.14 (0.38) | 0.23 (0.21) |
| Asian American | 0.27 (0.28) | 0.14 (0.32) |
| Other race | 0.36 (0.47) | 0.11 (0.28) |
| Protestant | 0.15 (0.27) | -0.32 (0.16) |
| Other religion | -1.23† (0.67) | -0.26 (0.29) |
| No religion | -0.50* (0.25) | -0.01 (0.27) |
| Parental annual income | -0.06 (0.05) | 0.01 (0.03) |
| Parental highest degree | -0.16 (0.11) | 0.01 (0.10) |
| Average similarity (Peer influence effect) | 2.20 (1.76) | 3.87 (3.09) |
| In-degree centrality | -0.07† (0.04) | 0.01 (0.03) |
| Extraversion | 0.18 (0.13) | -0.02 (0.11) |
| Agreeableness | 0.07 (0.17) | -0.01 (0.11) |
| Conscientiousness | 0.14 (0.15) | 0.02 (0.08) |
| Neuroticism | -0.20 (0.16) | -0.12 (0.10) |
| Openness | -0.04 (0.15) | -0.11 (0.13) |
| Frequency of discussing politics with friends | 0.15† (0.08) | -0.06 (0.06) |
| Democrat | -2.52*** (0.44) | -1.03*** (0.25) |
| Independent | -0.71* (0.33) | -0.47** (0.17) |
| Overall maximum convergence ratio | 0.24 | 0.23 |

† Two-sided $p<0.1$

* Two-sided $p<0.05$

** Two-sided $p<0.01$

*** Two-sided $p<0.001$.

Partisan affiliation is a strong predictor of changes in political orientation. Compared to NetSense study participants who voted for Obama or other candidates, those who chose Mitt Romney during the presidential election 2012 were about 12.4 times [exp(2.52)] and 2.0 times [exp(0.71)] more likely to become more conservative, respectively. Partisan affiliation was also a strong predictor of changes in self-placement in the NetHealth cohort. Compared with NetHealth study participants who considered themselves Republicans, the probabilities of increasing one unit on the liberal-conservative spectrum were 35.7% [1/exp(1.03)] for those who regard themselves as Democrats and 62.5% [1/exp(0.47)] for those who regard themselves as Independents. Overall, this shows that the partisan affiliation students bring with them impacts changes in their political orientations if and when they occur. To rule out the possibility that the strong correlation between political orientation and partisanship could impact other parameter estimates, we estimate an ancillary Model 2a that excludes partisanship from the political orientation equation for the NetHealth study participants. As shown in S2 Table,

**Table 7. The network equation in estimated SAO models for the NetSense and NetHealth study participants.**

|  | NetSense | NetHealth |
|---|---|---|
|  | Model 1 | Model 2 |
|  | beta (s.e.) | beta (s.e.) |
| Rate parameter (period 1) | 12.93*** (1.04) | 41.51*** (5.84) |
| Rate parameter (period 2) | 9.25*** (0.62) | 37.96*** (5.48) |
| Rate parameter (period 3) | 6.02*** (0.57) | 24.49*** (2.43) |
| Out-degree (density) | -6.27*** (0.34) | -7.16*** (0.38) |
| Reciprocityss | 7.25*** (0.70) | 10.35*** (0.62) |
| Transitive triplets | 0.92*** (0.10) | 0.51*** (0.11) |
| Transitive reciprocated triplets | -0.91*** (0.13) | -0.32* (0.15) |
| Out-degree—activity | 0.17 (0.25) | -0.49* (0.23) |
| In-degree—activity | -0.26 (0.57) | 0.68 (0.42) |
| In-degree—popularity | -0.03 (0.05) | 0.00 (0.01) |
| In-in degree^(1/2) assortativity | 0.05 (0.12) | 0.27*** (0.05) |
| Same residence hall | 0.69*** (0.09) | 0.00 (0.10) |
| Woman alter | -0.02 (0.12) | 0.69** (0.26) |
| Woman ego | 0.02 (0.11) | -0.93* (0.36) |
| Gender homophily selection | -0.05 (0.07) | 0.17** (0.06) |
| Same race | 0.21** (0.07) | 0.14* (0.06) |
| Same religious preference | 0.14* (0.07) | 0.04 (0.04) |
| Parental annual income homophily selection | 0.28* (0.14) | 0.11 (0.09) |
| Parental highest degree homophily selection | -0.08 (0.15) | 0.04 (0.10) |
| Extraversion homophily selection | 0.05 (0.18) | 0.08 (0.14) |
| Agreeableness homophily selection | 0.19 (0.19) | 0.12 (0.12) |
| Conscientiousness homophily selection | -0.27 (0.19) | 0.15 (0.14) |
| Neuroticism homophily selection | 0.21 (0.20) | 0.17 (0.18) |
| Openness homophily selection | 0.26 (0.19) | 0.32* (0.14) |
| Frequency discussing politics with friends homophily selection | 0.19 (0.13) | 0.07 (0.07) |
| Same partisanship | 0.03 (0.08) | -0.04 (0.04) |
| Political orientation alter | -0.04 (0.05) | 0.09 (0.15) |
| Political orientation ego | 0.03 (0.05) | -0.09 (0.16) |
| Political orientation homophily selection | 0.14 (0.22) | 0.30 (0.18) |
| Overall maximum convergence ratio | 0.24 | 0.23 |

† Two-sided $p < 0.1$

* Two-sided $p < 0.05$

** Two-sided $p < 0.01$

*** Two-sided $p < 0.001$.

the statistically significance pattern of rest effects are exactly the same between Model 2 and Model 2a.

**Network selection dynamics.** The key finding in the network selection equation is the null effect of similarity in political orientation between a pair of ego and alter in either study. In addition, being liberal or conservative does not make ego a more or less attractive partner, nor does it make alters more or less likely to be chosen by ego. Overall, the dynamics of network change in the NetSense and NetHealth data set were largely independent of political orientation, suggesting that this was not a salient marker governing tie formation dynamics in this population.

In terms of additional network effects, rate parameters from the network equation of Models 1 and 2 suggest NetHealth study participants had higher rates of turnover in their personal networks than NetSense study participants, resulting from the lower Jaccard indices shown in Table 5 and larger sample size. With regard to the endogenous structural effects at the dyadic level, the significantly negative out-degree (density) parameter indicates that both NetSense and NetHealth study participants were unlikely to connect with arbitrary others. The positive and statistically significant reciprocity parameter indicates that they tended to retain ties that had already connected with them at the previous time point.

Endogenous structural effects at the triadic level are similar across the two cohorts. With the transitive reciprocated triplets effect being adjusted for as suggested by Block [58], both NetSense and NetHealth study participants showed a tendency to form transitive triads (i.e., to connect with a current associate's associate). At the higher network level, NetHealth study participants who communicate with many associates are unlikely to form new ties and they are inclined to connect to those with similar in-degree centrality.

As for homophily beyond political orientation, the only statistically significant effect shared in common for NetSense and NetHealth study participants is that they were both prone to connect with others of the same racial identification, suggesting that this is indeed a salient axis for association in this population as it is for most Americans [59]. NetSense study participants were more likely to link others living in the same residence hall, having the same religious preference, and having similar parental annual income. Among NetHealth study participants, women initiated fewer ties and received more ties than men, and we also observed a high propensity to form same-gender ties [60]. We also see some evidence of network sorting by personality traits, namely trait openness to experience in this cohort.

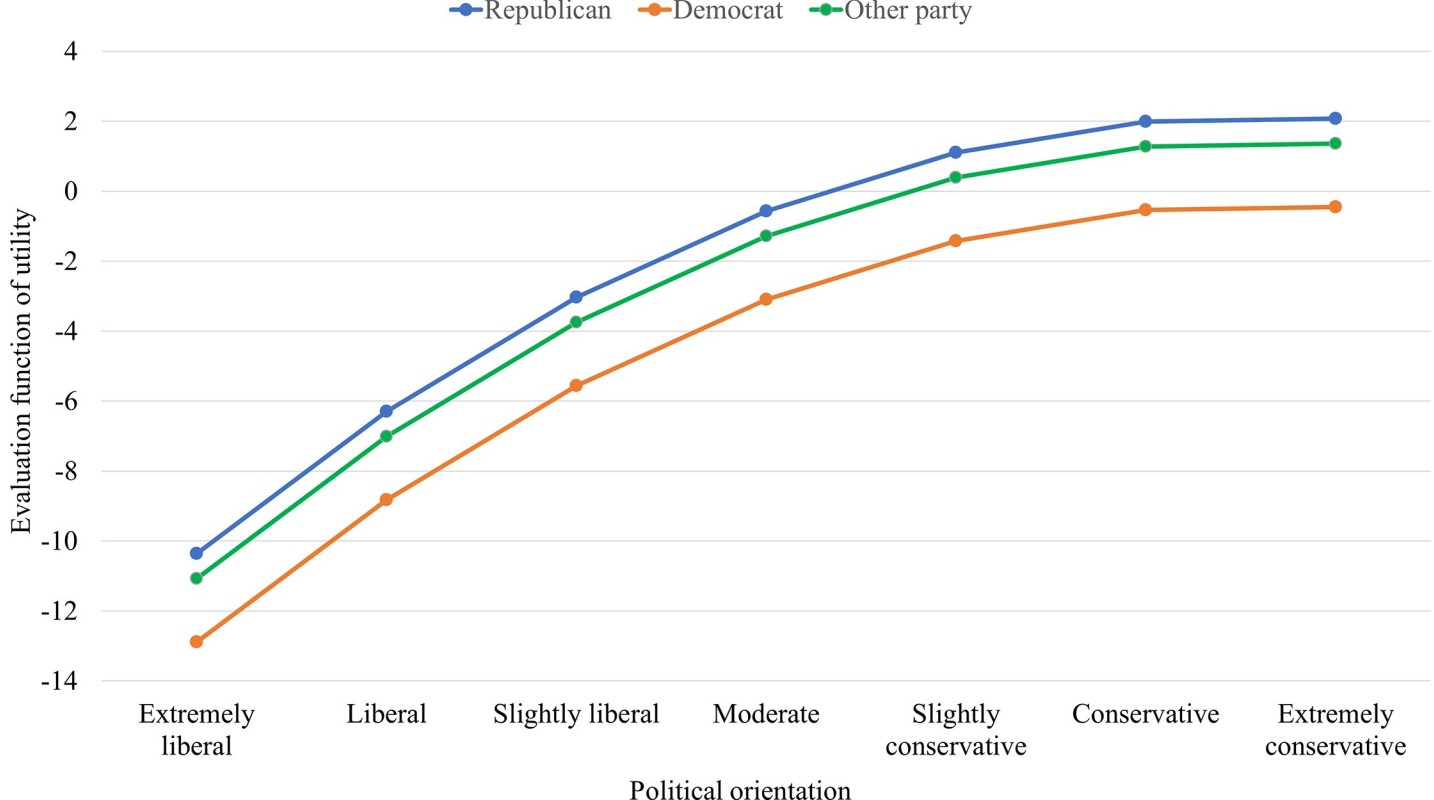

**Fig 1. Evaluation function of utility from linear shape, quadratic shape, and partisanship effects to political orientation among NetSense study participants.**

## Discussion

### Summary of findings

**Temporal trends in political orientation.** The current study examines the co-evolution of an individual's political orientation and his or her social networks using longitudinal data from two University of Notre Dame undergraduate student cohorts. The overall mean level of political orientation remained fairly constant as each cohort aged, and the differences between cohorts are for the most part minimal. At the individual level, there is also a good deal of stability in students' political orientation though there is some individual level change which can be modeled. While NetSense study participants became slightly more conservative over their first four semesters at Notre Dame, a finding that is in line with Bettencourt et al. [6], NetHealth study participants did not show this tendency.

**Partisanship.** Unsurprisingly, partisanship turned out to be a strong determinant of changes in political orientation, with Democrats and Independents being less likely to become more conservative than Republicans. This finding is also consistent with previous literature [1, 44, 47]. As we discussed earlier, there are two schools of thought about whether Americans are segmented and polarized along partisan lines. Our findings provided corroborative evidence for both positions. On the one hand, we find some evidence for greater polarization in the finding that partisanship predicts changes in political orientation. A pre-existing group identity, party affiliation, shapes how people change their political orientations when they do change, with Democrats and Independents more likely to become more liberal and Republicans more conservative. On the other hand, contrary to polarization ideas, we find no evidence of network sorting based on pre-existing political orientation and party affiliation. As students transition to college and build their personal networks, participants in both studies link to others without regard to whether they have similar or different partisanship or political orientations. This suggests that, net of other factors, the social ties the students form are not increasing polarization by creating an "echo chamber" in which people are connected to only those holding similar views.

**Socio-demographic factors.** As in prior research, demographic factors such as gender [6, 10], socioeconomic factors such as parental income and education [10, 61], and cognitive factors such as extraversion, agreeableness, and neuroticism [3] were not found to affect changes in political orientation. NetSense study participants who prefer a non-Christian religion or no religion tended to be less likely to become more conservative than their Catholic and Protestant counterparts, but this effect was not detected among NetHealth study participants. Therefore, the findings related to religious preference are mixed and thus reflective of previous work [12–14].

**Other factors.** Changes in political orientation were not influenced by other factors including racial identity, peers, popularity, cognitive factors such as conscientiousness and openness, and frequency of discussing politics with friends. Among previous research which argued that those factors predict political orientation, some relied on impressionistic or case-based observations to justify their claim [62], some failed to conduct a longitudinal study [3, 63], some had small sample sizes [17, 64], some focused on political deviants instead of regular people [63, 65], and none of them controlled for all above-mentioned factors in a single modeling framework as in the current study. Those reasons might account for some of the discrepancies between our findings and theirs.

**Network/Politics coevolution.** Findings from the network coevolution model suggest, in line with Lazer et al. [17], that positive selection into social relationships based on similarity in political orientation is *not* a significant factor driving network changes. We also find no evidence that other political factors (e.g., partisanship, and frequency of discussing politics with

friends) play a role in tie formation via positive self-selection of individuals who are similar in these factors into social ties. So while there is differentiation in terms of political orientation, it has not resulted in segmentation among students into distinct social groups based on political views with few cross-cutting ties.

Instead, we observe that changes in network ties are basically governed by endogenous network structural effects and racial homophily, the last result confirming this factor as a primary one governing social interaction in the American population [59]. NetSense study participants are more likely to form ties with others living in the same residence hall (a propinquity effect), having the same religious preference, and having similar parental income. NetHealth study participants are more likely to form ties with others of the same gender and similar levels of trait openness.

Importantly, we found no evidence for inter-personal social influence. The similarity of a person's political orientation to those of their friends does not predict changes in people's political orientation. Together with the finding that similarity in political orientation does not predict tie formation or persistence, there is no evidence that the social networks that form within a cohort of college students and the influence processes that could result from interaction within those networks are increasing polarization in political orientation. Instead, the main mechanism we identified in the change process is a relatively stable group identity, political party affiliation, that through directing how change occurs amplifies political orientation differences.

To summarize, while we find that there are cross-individual differences in political orientation and partisanship identification within the two cohorts, these differences are not accompanied by two traits of a polarized community: lack of cross-cutting ties and movement towards the extremes. Moreover, these differences fail to predict tie formation, a necessary requirement for group polarization. Cross-cutting ties between people with different political orientations do form, providing an avenue for social influence and minimizing the possibility of "echo chamber" like social networks reinforcing political orientations [42, 46]. We also fail to find evidence for participants in either study to move to the extremes of the liberal-conservative dimension. There is a high degree of stability in political orientation, and the change that does occur tends to be relatively small in scale. Finally, we find little evidence of social influence whereby people adjust their political orientation to match that of others they are connected to; instead, it seems that the change that does occur happens independently of (direct) interpersonal social influence. What we observe is that group identity, in the form of political party affiliation, informs the direction of change with Republicans becoming more conservative and Democrats and Independents more liberal over time, a finding that is consistent with recent work on the role of identity in the organization of political attitudes [66].

## Limitations, implications and suggestions for future research

There are some limitations to the current study that are worth noting. First, the data of both cohorts came from a private university predominantly comprised of white Catholic students. While our models adjust for the effects of racial identification and religious preference, generalizing of our findings to the broader population needs to be done with care. Future work should strive to replicate our approach using more representative samples containing longitudinal data on networks and political orientation. Second, information on family members' especially parents' political orientation and partisanship was not collected in either NetSense or NetHealth data, which means that we cannot look at intergenerational transmission processes. Parents' political orientation might be a good predictor for that of their children [67, 68], although in a more recent study parents' political orientation was not found to affect their

children's attitude toward feminism [6]. Future work should examine the role of family influences when studying the co-evolution of one's political orientation and his or her social networks. Finally, we did not know how each NetSense and NetHealth study participant engaged in politics more broadly. Political engagement could act as a moderator of the associations between religiosity and political attitudes [14]. It is worthwhile to include this item in future work on the subject.

Despite these limitations, the current study has important implications for the literature on political networks. First, this study shows both feasibility and merit of using a single modeling framework to study the co-evolution of political orientation and social networks. Such a model reduces the risk of over-estimating the operation of mutually correlated mechanisms (e.g., selection and peer influence). The SAOM approach minimizes the possibility of spurious association between political orientation and network dynamics with other factors while modelling network change and behavioral change in a unified way. This strategy also clearly distinguishes the effects of covariates and background factors on tie formation processes from the effects of same factors on changes in behavior and attitudes. Additionally, fitting our models to multiple data sources collected using the same design allowed us to access how and when change in political orientation occurs in a key period of the life-course. Our results are consistent with previous work finding that political orientations, at least among young adults, are not (yet) a strong factor canalizing social interaction and tie formation toward like-minded groups. Instead, we show that the changes in political orientation that do occur are more a function of "inertia" exerted by pre-existing identities operating at a slower pace (in our case party affiliation), than social influence or network selection processes at shorter time-scales [69].

## Supporting information

**S1 Table. Summary statistics of gender, racial identification, and religious factors among NetSense and NetHealth study participants who stayed and dropped out study.**
(DOCX)

**S2 Table. The estimated SAO model excluding partisanship in the political orientation equation for the NetHealth study participants.**
(DOCX)

**S1 File. Data for the NetSense study participants.**
(ZIP)

## Author Contributions

**Conceptualization:** Cheng Wang.

**Data curation:** Cheng Wang.

**Formal analysis:** Cheng Wang.

**Methodology:** Cheng Wang.

**Project administration:** Omar Lizardo.

**Software:** Cheng Wang.

**Supervision:** Omar Lizardo.

**Validation:** David S. Hachen.

**Visualization:** Cheng Wang.

**Writing – original draft:** Cheng Wang.

**Writing – review & editing:** Cheng Wang, Omar Lizardo, David S. Hachen.

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
