## [Decision Letter · Decision Letter 0]

31 Jan 2020

PONE-D-19-35811

Neither Influence Nor Selection: Examining Co-evolution of Political Orientation and Social Networks in the NetSense and NetHealth Studies

PLOS ONE

Dear Dr. Wang,

Thank you for submitting your manuscript to PLOS ONE. After careful consideration, we feel that it has merit but does not fully meet PLOS ONE’s publication criteria as it currently stands. Therefore, we invite you to submit a revised version of the manuscript that addresses the points raised during the review process.

We would appreciate receiving your revised manuscript by Mar 16 2020 11:59PM. To enhance the reproducibility of your results, we recommend that if applicable you deposit your laboratory protocols in protocols.io, where a protocol can be assigned its own identifier (DOI) such that it can be cited independently in the future. For instructions see: http://journals.plos.org/plosone/s/submission-guidelines#loc-laboratory-protocols

We look forward to receiving your revised manuscript.

Kind regards,

Valerio Capraro

Academic Editor

PLOS ONE

Additional Editor Comments (if provided):

I have now collected three reviews from three experts in the field. All reviewers find the study interesting and well executed, but suggest several improvements. Please address they comments at your earliest convenience.

I am looking forward for the revision.

Journal Requirements:

Reviewers' comments:

Reviewer's Responses to Questions

**Comments to the Author**

1. Is the manuscript technically sound, and do the data support the conclusions?

Reviewer #1: Partly

Reviewer #2: Partly

Reviewer #3: Partly

2. Has the statistical analysis been performed appropriately and rigorously? 

Reviewer #1: Yes

Reviewer #2: I Don't Know

Reviewer #3: Yes

3. Have the authors made all data underlying the findings in their manuscript fully available?

Reviewer #1: No

Reviewer #2: Yes

Reviewer #3: No

4. Is the manuscript presented in an intelligible fashion and written in standard English?

Reviewer #1: Yes

Reviewer #2: Yes

Reviewer #3: Yes

5. Review Comments to the Author

Reviewer #1: Summary:

The paper reviews the factors responsible for changes in political orientation, including network effects. The paper focuses on a student sample at Notre Dame University. For the analysis, the paper uses a panel setup (4 semesters) with stochastic actor-based models (SABM). The paper finds a general trend towards conservatism, but limited evidence for peer effects.

Major Comments:

1) I party disagree with the interpretation of the paper results that we can rule out the influence of social networks. Arguably, there is some evidence of peer effects on political orientation, even if the statistical significance is only at 0.1 (Table 6, Model 2). However, an alternative explanation is that the weak statistical significance may be partly due to the small sample size and the fact that political orientation is relatively stable (page 13).

Also, it is unclear what positive evidence there is that embeddedness in larger institutional contexts may explain the shift towards a more conservation political orientation among the students. Rather, the paper results are also consistent with an explanation that more people in the US became more conservative over the time period, including the students.

2) Partisanship - as noted in the paper, there is a very high correlation between political orientation and partisanship (page 6, above 0.90). Thus, including partisanship in the SABM models may introduce multicollinearity. As well, there are concerns of post-treatment bias. If people who become more conservation also become more republican, controlling for partisanship could diminish the estimate size.

To account for these concerns, I would suggest running a SABM model that does not control for partisanship.

Minor comments:

3) According to Table 1, there seems to be a lot of attrition in the sample, partly for technical reasons. Thus, it would be nice to see some analysis that rules out some sort of attrition bias (page 9). This can be done by showing a balance table for respondents who did (or did not) drop out of the sample.

4) Measurement concerns – I would like to see more details on how call detail records were used to measure one’s network. Also, it was not entirely clear how this measure related to the residence hall.

5) There is a lot of technical language in the paper. To make the paper more accessible to a broader audience, I would suggest giving more explanation about certain terms (such as ego or alter), possibly in the footnotes.

Reviewer #2: 1. The network members were samples from their cohorts. This is not a very good network delineation. However, these models will give meaningful results for sampled data in the sense that it may be assumed that signs of parameters will be preserved.

2. The paper is not clear about the network data collection. How were the data coded, i.e., when was a tie supposed to have been observed?

3. If the partisanship is a changing variable, it potentially takes away much or all of the meaningful variability in political orientation. Therefore, if changing, it should be used either as a co-evolving dependent variable, or only the wave-1 value should be used. If my understanding is correct, it is non-changing for NetSense and changing for NetHealth.

4. An important interpretation of SABM parameters is that they do not directly reflect differences in change. In this case, since Democrats on average have lower values on the political orientation scale to begin with, their partisanship needs to have a negative parameter in order to keep them at lower values. Therefore, the interpretation that they are “more likely to trend conservative” (p. 17) is incorrectly stated. The curves in Figure 1 could be differentiated for Republicans and Democrats and then are two quadratic functions shifted with respect to each other, the Democrats’ curve being more to the left. Such curves will eventually lead to an equilibrium; when reached, there would be on average no change, but Democrats would fluctuate around a less conservative mean value than Republicans.

5. I am very surprised at the results for the NetHealth group, so much so that I doubt them. Given the Jaccard index values, and the adequate fit as mentioned (very briefly, without any details), I would not expect to have rate functions this high; and with small standard errors, too. Also I would not expect negative transitivity and positive 3-cycle effects. This might be related to the incompleteness of the network delineation; but I rather suspect some error. Is the analysis reproducible, i.e., can the data be made available?

6. The SES measure is defined with equidistant boundaries on a dollar scale. Given the nature of money, other class boundaries, growing broader for higher values, might be better (e.g., equidistant logarithms or nearly so).

7. For the goodness of fit test (Monte Carlo test of Mahalanobis Distance), the appropriate reference is

Lospinoso, Josh, and Tom A.B. Snijders (2019), Goodness of Fit for Stochastic Actor-Oriented Models. Methodological Innovations, 12 (September 2019). doi:10.1177/2059799119884282.

8. Please give percentages in Tables with no decimals; these make no sense (just think about the number of persons corresponding to the second decimal…).

9. p. 13 line 4: … divided by all ties existing at least in one of the consecutive waves ….

10. Since the microsteps are part of the model assumptions and not of the results, it might be better to mention and briefly describe them earlier than on p. 16.

11. There are some language issues, e.g., on p. 17, “with NetSense participants … became” and “The estimate parameter”.

Reviewer #3: I think the paper is almost ready for publication in Plos One. However, I have one concern and a few suggestions about the SAOM/SABM model specification, in particular about the selection model.

Main concern: An outdegree-activity effect is missing from the model. This is necessary to include in order to control for different nomination activities of individuals.

Further suggestions:

1. I would also recommend controlling for indegree-activity or outdegree-popularity in order to model the relationship between the indegree and the outdegree distributions - given that the authors seem to have sufficient data for this.

2. The authors use the collection of reciprocity, transitive triplets, 3-cycles and transitive reciprocated triplets. Theoretically, either 3-cycles or the transitive reciprocated triplets effect is sufficient, and the two together may be superfluous. In Block (2015; cited by the authors), transitive reciprocated triplets are proposed as an alternative of the 3-cycle effect (even though models with both effects are also estimated in that article). In line with most published SAOM articles, I would recommend that the authors to use either 3-cycles or transitive reciprocated triplets in their models.

3. Unless there is a specific reason not to do so, I would also recommend that the authors use the term stochastic actor-oriented models (SAOM) instead of stochastic actor-based models (SABM), given that in recent years, this latter name has been used predominantly in literature.

6. PLOS authors have the option to publish the peer review history of their article (what does this mean?). If published, this will include your full peer review and any attached files.

Reviewer #1: Yes: Michael Freedman

Reviewer #2: No

Reviewer #3: No

---

## [Author Response · Author response to Decision Letter 0]

13 Mar 2020

Response to Reviewers

PONE-D-19-35811

Neither Influence Nor Selection: Examining Co-evolution of Political Orientation and Social Networks in the NetSense and NetHealth Studies

PLOS ONE

We would like to thank the reviewers for their very constructive feedback. Incorporating the reviewer suggestions for revision has resulted in a greatly improved manuscript. Below, we note the concerns of the reviewers and then explain how we responded to each comment. In all cases, reviewer comments are in italics, while our own comments are in plain text.

Reviewer 1

Major Comments:

1) I party disagree with the interpretation of the paper results that we can rule out the influence of social networks. Arguably, there is some evidence of peer effects on political orientation, even if the statistical significance is only at 0.1 (Table 6, Model 2). However, an alternative explanation is that the weak statistical significance may be partly due to the small sample size and the fact that political orientation is relatively stable (page 13). 

––––––––––––––––––––––––––––––––––––––––––––––––––––

We appreciate the reviewer’s concern with regard to the possible underestimation of influence effects. Following the suggestions of reviewer 3, we now include an out-degree - activity effect and an in-degree - activity effect and remove 3-cycles effect from the network equation. In the new model, the peer influence effect on political orientation is no longer statistically significant at the 0.1 level. This indicates that we were actually slightly over-estimating influence effects in previous specifications. We specified this model with the most recent version of RSiena software package (1.2-24 released on February 16, 2020) 30 times and the result is consistent. We conclude therefore that in these cohorts there is no social influence.

Also, it is unclear what positive evidence there is that embeddedness in larger institutional contexts may explain the shift towards a more conservation political orientation among the students. Rather, the paper results are also consistent with an explanation that more people in the US became more conservative over the time period, including the students.

––––––––––––––––––––––––––––––––––––––––––––––––––––

The concern from the reviewer is reasonable, given we only have data from one private university predominantly comprised of white Catholic students. We cannot rule out the possibility that this larger institutional context is driving its students towards a more conservative direction. But we do adjust for racial identification and religious factors in both models. And we concede this limitation on page 25. We expect future work to replicate our approach using more representative samples containing longitudinal data on networks and political orientation.

Moreover, we now focus less on emphasizing the large institutional context explanation of changes in political orientation and more on the importance of a group affiliation and identify, partisanship and political party identity.

2) Partisanship - as noted in the paper, there is a very high correlation between political orientation and partisanship (page 6, above 0.90). Thus, including partisanship in the SABM models may introduce multicollinearity. As well, there are concerns of post-treatment bias. If people who become more conservation also become more republican, controlling for

partisanship could diminish the estimate size. To account for these concerns, I would suggest running a SABM model that does not control for partisanship.

––––––––––––––––––––––––––––––––––––––––––––––––––––

We understand the reviewer’s concern and estimate an ancillary Model 2a that excludes the partisanship from the political orientation equation for the NetHealth study participants. As shown in S2 Table, the patterns of statistical significance in the rest of the effects are exactly the same between Model 2 and Model 2a. We now discuss this in the first paragraph on page 20.

Moreover, the RSiena software package includes a set of collinearity check routines. First, multicollinearity can be detected in the covariance matrix. Second, we would see very large standard errors for coefficient estimates of variables with multicollinearity. Third, when there is perfect correlation or near-perfect correlation between two variables, the RSiena software package will automatically stop and show a series of warning messages. We did not encounter any such issues when estimating Model 2, suggesting that multicollinearity is not a major cause of concern.

Minor comments:

3) According to Table 1, there seems to be a lot of attrition in the sample, partly for technical reasons. Thus, it would be nice to see some analysis that rules out some sort of attrition bias (page 9). This can be done by showing a balance table for respondents who did (or did not) drop out of the sample.

––––––––––––––––––––––––––––––––––––––––––––––––––––

We now add an S1 Table displays the summary statistics of gender, racial identification, and religious preference among NetSense and NetHealth study participants, which are generally similar between stayers and dropouts from wave 2 to wave 4. To address sample selection concerns, we also ran additional logistic regression models predicting the dropout status of NetSense and NetHealth study participants with these factors. The only statistically significant effect is that a NetHealth study participant identifying as the “other” race is 127% more likely to drop out of the study than a white participant at wave 4 (p < 0.05). We now discuss this in the “Sample attrition” section on pages 14-15.

4) Measurement concerns – I would like to see more details on how call detail records were used to measure one’s network. 

––––––––––––––––––––––––––––––––––––––––––––––––––––

On pages 10-11 we now add: “The raw CDRs are used to construct an adjacency matrix representing a communication network (based on text and calls) for each semester. The adjacency matrix was built based on communication events in the semester of the corresponding survey from which we obtained political orientation data was administered. A directed tie between person i and person j exists in a given semester if i initiated a communication event (call or text) with j during that semester. Each adjacency matrix is asymmetric above and below the diagonal, given that communication relationships between an “ego” (i.e. a focal study participant) and his or her “alter” (i.e. another participant to whom ego is directly connected) is a directed tie and might not always be reciprocal. Since these network ties are between with-study participants, we do not use a cutoff on tie strength, i.e., the frequency or duration of communicant events.”

Also, it was not entirely clear how this measure related to the residence hall.

––––––––––––––––––––––––––––––––––––––––––––––––––––

On page 12 we now add: “Given participants living in the same residence halls could be more likely to establish relationships among themselves than with those living in different residence halls, we use the commonality of their residence halls to predict the formation and maintenance of social network ties among them.”

5) There is a lot of technical language in the paper. To make the paper more accessible to a broader audience, I would suggest giving more explanation about certain terms (such as ego or alter), possibly in the footnotes.

––––––––––––––––––––––––––––––––––––––––––––––––––––

We thank the reviewer for making this suggestion. We now explain ego as “a focal study participant” and alters as “another participant to whom ego is directly connected” on page 11. We also explain what certain terms mean at first used and try to minimize the use of technical terms throughout.

Reviewer 2

1. The network members were samples from their cohorts. This is not a very good network delineation. However, these models will give meaningful results for sampled data in the sense that it may be assumed that signs of parameters will be preserved.

––––––––––––––––––––––––––––––––––––––––––––––––––––

It is true that the NetSense and NetHealth cohorts are a subset of the members of their respective college cohorts. Compared to the full cohort they are drawn from, their distribution demographic and family background traits are very similar to the “typical” incoming Notre Dame student. While we do have information on our participants ties to people outside the sample (i.e. family, other Notre Dame students, etc.) we restrict all analysis to ties formed between participants in the study. While we do not have all the ties between students in each of these cohorts, we find no evidence that the ties we do have are different from other ties within each college student cohort where one of the person’s is not in the sample.

2. The paper is not clear about the network data collection. How were the data coded, i.e., when was a tie supposed to have been observed?

––––––––––––––––––––––––––––––––––––––––––––––––––––

On pages 10-11 we now add: “The raw CDRs are used to construct an adjacency matrix representing a communication network (based on text and calls) for each semester. The adjacency matrix was built based on communication events in the semester of the corresponding survey from which we obtained political orientation data was administered. A directed tie between person i and person j exists in a given semester if i initiated a communication event (call or text) with j during that semester. Each adjacency matrix is asymmetric above and below the diagonal, given that communication relationships between an “ego” (i.e. a focal study participant) and his or her “alter” (i.e. another participant to whom ego is directly connected) is a directed tie and might not always be reciprocal. Since these network ties are between with-study participants, we do not use a cutoff on tie strength, i.e., the frequency or duration of communicant events.”

3. If the partisanship is a changing variable, it potentially takes away much or all of the meaningful variability in political orientation. Therefore, if changing, it should be used either as a co-evolving dependent variable, or only the wave-1 value should be used. If my understanding is correct, it is non-changing for NetSense and changing for NetHealth.

––––––––––––––––––––––––––––––––––––––––––––––––––––

The reviewer is right that the partisanship is non-changing for the NetSense study participants but changing for the NetHealth study participants. Given the high stability of partisanship status among the NetHealth study participants, now we follow the reviewer’s suggestion and use only the wave-1 value in the Model 2 for the NetHealth study participants. And we clarify this in the second paragraph on page 9.

4. An important interpretation of SABM parameters is that they do not directly reflect differences in change. In this case, since Democrats on average have lower values on the political orientation scale to begin with, their partisanship needs to have a negative parameter in order to keep them at lower values. Therefore, the interpretation that they are “more likely to trend conservative” (p. 17) is incorrectly stated. The curves in Figure 1 could be differentiated for Republicans and Democrats and then are two quadratic functions shifted with respect to each other, the Democrats’ curve being more to the left. Such curves will eventually lead to an equilibrium; when reached, there would be on average no change, but Democrats would fluctuate around a less conservative mean value than Republicans.

––––––––––––––––––––––––––––––––––––––––––––––––––––

We thank the reviewer for this astute comment. In response, we now add the partisanship effect to the evaluation function of utility for political orientation in revised Fig 1. As the reviewer anticipated, NetSense study participants who voted as Democrats or Independents do have lower values of utility than those voted as Republicans. However, the curve lines for three types of partisanship status are isomorphic along the Y axis instead of the X axis. Therefore, our statement that the NetSense study participants are likely to trend conservative still holds, even for those who voted as Democrats. 

For example, consider a NetSense study participant who voted as Democrat has an “Extremely liberal” position at wave 1. At the next time point he or she has two choices, staying in the same position or changing to “Liberal”. Based on our model, as well as shown in the revised Fig 1, he or she is more likely to select the latter choice because of the higher utility value. This decision is without regard to his or her partisanship status.

Due to the negative quadratic shape effect, the equilibrium status appears at the right side of three curve lines when they become relatively flat, i.e., the utility change between the “Conservative” and “Extremely conservative” positions is much smaller than that between the “Conservative” and Slightly conservative” positions. In other words, there is not much incentive for the NetSense study participants to change their political orientation from “Conservative” to “Extremely conservative” when compared to changes between other adjacent positions.

5. I am very surprised at the results for the NetHealth group, so much so that I doubt them. Given the Jaccard index values, and the adequate fit as mentioned (very briefly, without any details), I would not expect to have rate functions this high; and with small standard errors, too. 

––––––––––––––––––––––––––––––––––––––––––––––––––––

According to Table 5, the NetHealth cohort has a lower Jaccard index value than the NetSense cohort, indicating lower network stability. Moreover, the sample size of the NetHealth group is 483 and that for the NetSense group is 196. Therefore, it makes sense that a NetHealth study participant has higher turnover count with his or her associates. And this is reflected by the larger parameter values and standard errors for the rate function in Model 2 for the NetHealth study participants. We now discuss this on pages 20-21.

Also I would not expect negative transitivity and positive 3-cycle effects. This might be related to the incompleteness of the network delineation; but I rather suspect some error. 

––––––––––––––––––––––––––––––––––––––––––––––––––––

Following the suggestions of reviewer 3, we now include an outdegree-activity effect and in-degree activity effect and remove 3-cycles effect from the network equation. In the new models, the transitivity effect is now positive.

Is the analysis reproducible, i.e., can the data be made available?

––––––––––––––––––––––––––––––––––––––––––––––––––––

We attach NetSense study data as S1 File. The NetHealth data are now publicly online at http://sites.nd.edu/nethealth.

6. The SES measure is defined with equidistant boundaries on a dollar scale. Given the nature of money, other class boundaries, growing broader for higher values, might be better (e.g., equidistant logarithms or nearly so).

––––––––––––––––––––––––––––––––––––––––––––––––––––

As shown in Table 3, parental annual income has a broad range for higher values. While ideally we would like to check the robustness of the findings using other cutpoints for the categories, the survey questions asked students to select which of these categories characterized their parent’s income, so we do not have access to a different income measure. 

7. For the goodness of fit test (Monte Carlo test of Mahalanobis Distance), the appropriate reference is Lospinoso, Josh, and Tom A.B. Snijders (2019), Goodness of Fit for Stochastic Actor-Oriented Models. Methodological Innovations, 12 (September 2019). doi:10.1177/2059799119884282.

––––––––––––––––––––––––––––––––––––––––––––––––––––

We have now added this reference in the revised manuscript.

8. Please give percentages in Tables with no decimals; these make no sense (just think about the number of persons corresponding to the second decimal…).

––––––––––––––––––––––––––––––––––––––––––––––––––––

We now follow the reviewer’s suggestion and give percentages with no decimals in all tables. 

9. p. 13 line 4: … divided by all ties existing at least in one of the consecutive waves ….

––––––––––––––––––––––––––––––––––––––––––––––––––––

We thank the reviewer for raising this important point. We now modified the definition of Jaccard index as suggested by the reviewer.

10. Since the microsteps are part of the model assumptions and not of the results, it might be better to mention and briefly describe them earlier than on p. 16.

––––––––––––––––––––––––––––––––––––––––––––––––––––

We thank the reviewer for this suggestion. After introducing the rate function in the second paragraph on page 13 we now add: “The model is then estimated by simulating the networks and behavior forward in time. Therefore, there are many microsteps in which actors update their evaluation functions by changing their states on political orientation and network tie choice (i.e., -1 unit, no change, or +1 unit).”

11. There are some language issues, e.g., on p. 17, “with NetSense participants … became” and “The estimate parameter”.

––––––––––––––––––––––––––––––––––––––––––––––––––––

We have fixed all these grammatical errors.

Reviewer 3

Main concern: An outdegree-activity effect is missing from the model. This is necessary to include in order to control for different nomination activities of individuals.

––––––––––––––––––––––––––––––––––––––––––––––––––––

This is a valuable suggestion. We now add the out-degree - activity effect in both models. 

1. I would also recommend controlling for indegree-activity or outdegree-popularity in order to model the relationship between the indegree and the outdegree distributions - given that the authors seem to have sufficient data for this. 

––––––––––––––––––––––––––––––––––––––––––––––––––––

We thank the reviewer for this suggestion. We now add the in-degree - activity effect in both models.

2. The authors use the collection of reciprocity, transitive triplets, 3-cycles and transitive reciprocated triplets. Theoretically, either 3-cycles or the transitive reciprocated triplets effect is sufficient, and the two together may be superfluous. In Block (2015; cited by the authors), transitive reciprocated triplets are proposed as an alternative of the 3-cycle effect (even though models with both effects are also estimated in that article). In line with most published SAOM

articles, I would recommend that the authors to use either 3-cycles or transitive reciprocated triplets in their models.

––––––––––––––––––––––––––––––––––––––––––––––––––––

We have now removed the 3-cycles effect from both models.

3. Unless there is a specific reason not to do so, I would also recommend that the authors use the term stochastic actor-oriented models (SAOM) instead of stochastic actor-based models (SABM), given that in recent years, this latter name has been used predominantly in literature.

––––––––––––––––––––––––––––––––––––––––––––––––––––

We now follow the reviewer’s suggestion and replace stochastic actor-based models (SABM) with stochastic actor-oriented models (SAOM) throughout the manuscript.

---

## [Decision Letter · Decision Letter 1]

14 Apr 2020

PONE-D-19-35811R1

Neither Influence Nor Selection: Examining Co-evolution of Political Orientation and Social Networks in the NetSense and NetHealth Studies

PLOS ONE

Dear Dr. Wang,

Thank you for submitting your manuscript to PLOS ONE. After careful consideration, we feel that it has merit but does not fully meet PLOS ONE’s publication criteria as it currently stands. Therefore, we invite you to submit a revised version of the manuscript that addresses the points raised during the review process.

We would appreciate receiving your revised manuscript by May 29 2020 11:59PM. To enhance the reproducibility of your results, we recommend that if applicable you deposit your laboratory protocols in protocols.io, where a protocol can be assigned its own identifier (DOI) such that it can be cited independently in the future. For instructions see: http://journals.plos.org/plosone/s/submission-guidelines#loc-laboratory-protocols

We look forward to receiving your revised manuscript.

Kind regards,

Valerio Capraro

Academic Editor

PLOS ONE

Additional Editor Comments (if provided):

One of the reviewers still has some minor comments. Please address these remaining comments at your earliest convenience. I am looking forward for the final version.

Reviewers' comments:

Reviewer's Responses to Questions

**Comments to the Author**

1. If the authors have adequately addressed your comments raised in a previous round of review and you feel that this manuscript is now acceptable for publication, you may indicate that here to bypass the “Comments to the Author” section, enter your conflict of interest statement in the “Confidential to Editor” section, and submit your "Accept" recommendation.

Reviewer #1: All comments have been addressed

Reviewer #2: All comments have been addressed

2. Is the manuscript technically sound, and do the data support the conclusions?

Reviewer #1: Yes

Reviewer #2: Yes

3. Has the statistical analysis been performed appropriately and rigorously? 

Reviewer #1: Yes

Reviewer #2: Yes

4. Have the authors made all data underlying the findings in their manuscript fully available?

Reviewer #1: Yes

Reviewer #2: Yes

5. Is the manuscript presented in an intelligible fashion and written in standard English?

Reviewer #1: Yes

Reviewer #2: Yes

6. Review Comments to the Author

Reviewer #1: (No Response)

Reviewer #2: My remarks have been addressed adequately.

I still would like to make two comments.

1. The response rates of 100% at the first surveys seem too good to be true; although not impossible. Which universe is used here as the totality? All freshmen?

2. P. 10 states “A directed tie between person i and person j exists in a given semester if i initiated a communication event (call or text) with j during that semester”. From Table 5, it can be concluded that the average degree dropped from about 4 to about 2 for NetSense and from just over 8 to just over 6 for NetHealth. How is this possible? These numbers do not tally with the description of the network data collection; or else the smartphones used for data collection are only a minor medium of contact for these students.

7. PLOS authors have the option to publish the peer review history of their article (what does this mean?). If published, this will include your full peer review and any attached files.

Reviewer #1: Yes: Michael Freedman

Reviewer #2: No

---

## [Author Response · Author response to Decision Letter 1]

4 May 2020

Response to Reviewers

PONE-D-19-35811R1

Neither Influence Nor Selection: Examining Co-evolution of Political Orientation and Social Networks in the NetSense and NetHealth Studies

PLOS ONE

We would like to thank the reviewers for their very constructive feedback. Incorporating the reviewer suggestions for revision has resulted in a greatly improved manuscript. Below, we note the concerns of the reviewers and then explain how we responded to each comment. In all cases, reviewer comments are in italics, while our own comments are in plain text.

Reviewer 2

1. 1. The response rates of 100% at the first surveys seem too good to be true; although not impossible. Which universe is used here as the totality? All freshmen?

––––––––––––––––––––––––––––––––––––––––––––––––––––

The response rate is the number of participants who complete the survey divided by the number of participants in the sample (Babbie 2016). The NetSense study recruited 196 participants and the NetHealth study recruited 483 participants. Therefore, denominators for the response rates are 196 for NetSense study and 483 for NetHealth study.

2. P. 10 states “A directed tie between person i and person j exists in a given semester if i initiated a communication event (call or text) with j during that semester”. From Table 5, it can be concluded that the average degree dropped from about 4 to about 2 for NetSense and from just over 8 to just over 6 for NetHealth. How is this possible? These numbers do not tally with the description of the network data collection; or else the smartphones used for data collection are only a minor medium of contact for these students.

––––––––––––––––––––––––––––––––––––––––––––––––––––

The reviewer is right that the average degree dropped over the first two years for both NetSense and NetHealth participants. However, these numbers reflect the real situation. First, when a freshman made the transition from high school to college in an exclusively residential campus like the University of Notre Dame, his or her personal social networks were dramatically restructured and lifelong relationships (both romantic and friendship) began to form. Student personal networks experienced a large amount of turnover during the first semester, and gradually stabilized over time, as indicated by the Jaccard indices shown in Table 5. Second, since each NetSense participant got a free smartphone and phone plan, it is reasonable to expect average degree to drop by about 49% (from 4 to 2) in their fourth semester, compared to NetHealth participants who brought their own smartphones and used their own plans. Third, the average degrees reported in Table 5 are based on social ties between in-study participants, which accounted for 10% to 12% of all communication events via smartphones for NetSense and NetHealth participants. And we found that the number of all communication events via smartphones also decreased over time, the pattern of which proportionally matches that between in-study participants. These numbers are consistent with previous work showing that while people can interact via face-to-face meetings, letters, landlines, smartphones, emails, Internet, and many other ways, the use of one medium for communication between individuals is strongly correlated with communication via other channels (Haythornthwaite, 2005; Onnela et al., 2007).

References:

Babbie, E. 2016. The Basics of Social Research (7th Edition). Boston, MA: Cengage Learning.

Haythornthwaite, C. (2005). Social networks and internet connectivity effects. Information, Communication & Society, 8(2), 125–147.

Onnela, J. P., Arbesman, S., Gonzalez, M. C., Barabási, A. L., & Christakis, N. A. (2011). Geographic constraints on social network groups. PLoS ONE, 6(4), e16939.

---

## [Editor Report · Decision Letter 2]

6 May 2020

Neither Influence Nor Selection: Examining Co-evolution of Political Orientation and Social Networks in the NetSense and NetHealth Studies

PONE-D-19-35811R2

Dear Dr. Wang,

We are pleased to inform you that your manuscript has been judged scientifically suitable for publication and will be formally accepted for publication once it complies with all outstanding technical requirements.

With kind regards,

Valerio Capraro

Academic Editor

PLOS ONE
---

## [Editor Report · Acceptance letter]

14 May 2020

PONE-D-19-35811R2 

Neither Influence Nor Selection: Examining Co-evolution of Political Orientation and Social Networks in the NetSense and NetHealth Studies 

Dear Dr. Wang:

I am pleased to inform you that your manuscript has been deemed suitable for publication in PLOS ONE. Congratulations! Your manuscript is now with our production department. 

With kind regards,

on behalf of

Dr. Valerio Capraro 

Academic Editor

PLOS ONE